# Triggering Magnets for Wiegand Sensors: Electrodeposited and Origami-Magnetized CoNiP Micro-Magnets

**DOI:** 10.3390/s23136043

**Published:** 2023-06-29

**Authors:** Ganesh Kotnana, Yun Cheng, Chiao-Chi Lin

**Affiliations:** 1Department of Physics, School of Advanced Sciences, VIT-AP University, Amaravati 522237, Andhra Pradesh, India; ganesh.kotnana@vitap.ac.in; 2Department of Materials Science and Engineering, Feng Chia University, Taichung 40724, Taiwan

**Keywords:** Wiegand sensor, electrodeposited magnets, microfabrication, origami magnetization, pole pieces, trigging field, Wiegand pulse

## Abstract

Miniature sensors are key components for applications in the Internet of Things (IoT), wireless sensor networks, autonomous vehicles, smart cities, and smart manufacturing. As a miniature and self-powered magnetic sensor, the Wiegand sensor possesses advantageous traits including changing-rate-independent output, low cost, and remarkable repeatability and reliability. A typical Wiegand sensor requires hard magnetic pole pieces that provide external fields for triggering voltage outputs that are called Wiegand pulses. However, the wire-shaped sensing element of Wiegand sensors is the critical issue that limits the design, selection, and adoption of the external triggering magnets. Currently, the widely used pole piece materials are rare-earth magnets. However, adopting rare-earth magnets brings strong stray fields, causing an electromagnetic interference (EMI) problem. In this study, patterned CoNiP hard magnets were electrodeposited on flexible substrates through microfabrication. Origami magnetization was utilized to control the resultant stray fields and thus the pole piece of CoNiP magnets can successfully trigger the output of the Wiegand pulse. In comparison, the output voltage of the triggered pulse acquired through the patterned CoNiP magnets is comparable to that acquired by using the rare-earth magnets. Furthermore, both the volume (and hence the weight) of the Wiegand sensor and the EMI issue can be significantly reduced and mitigated, respectively, by the CoNiP magnets.

## 1. Introduction

Sensing technologies based on the magnetic properties of materials have drawn considerable interest due to the advantages of being non-contact and having a low input power requirement, superior energy efficiency, high sensitivity, and robust reliability against adverse environments [1,2]. Among the sensors that utilize various magnetic properties of materials, Wiegand sensors have the advantage of being self-powered and a unique capability for simultaneous sensing and energy harvesting [3,4,5]. Wiegand sensors are composed of a Wiegand wire that possesses different magnetic properties at the surface layer and the core, a pick-up coil that senses the change in magnetic flux of the Wiegand wire for inducing an output voltage pulse, and an external source of magnetic field to trigger the magnetic reversal of the Wiegand wire [6,7]. Federico Iob et al. proposed low-power wireless-charging novel architecture based on Wiegand sensors [8]. Wiegand sensors were explored for many industrial applications such as mechanical energy harvesting for the railway industry [9], linear positioning systems [10], providing power for magneto-impedance sensors [11], vibration-based electricity generators [12], and batteryless hall sensors [13].

The pickup coil wound around the Wiegand wire is as shown in Figure 1. Essentially, when the external magnetic field is applied in the opposite direction to the magnetization of the wire, the magnetization of the soft layer will flip first. During this process, there is a change in magnetic flux around the Wiegand wire. This change in the magnetic flux will induce an electromotive force in the pickup coil that can be visualized as a voltage pulse, also known as a Wiegand pulse. Since the magnetization reversal of the soft layer in the Wiegand wire is rapid and steep, the magnetization jump in the Wiegand wire is usually called the large Barkhausen jump and is also known as the Wiegand effect. The pulse generated by the Wiegand sensor can then be used as a power supply for equipment without batteries.

The design and adoption of external triggering fields for the Wiegand sensor are limited by the wire-shaped sensing element of the Wiegand wire, which is a critical issue. Recently, Lin et al. thoroughly reviewed the distinctive Wiegand sensor and pointed out that the sophisticated design and fabrication of the triggering magnets are key to widening the application scenarios of Wiegand sensors [7]. Hard magnetic pole pieces that are able to provide uniform external fields to trigger a uniform and complete magnetization reversal of the soft layer in the Wiegand wire are essential to maximize the output voltages. Currently, rare-earth magnets are widely used as pole piece materials, but they bring about strong stray fields that lead to electromagnetic interference (EMI) problems. Moreover, the typical diameter of the Wiegand wire is 0.25 mm [14]. Using bulk magnets impairs the miniaturization capability of Wiegand sensors. There is an urgent need to investigate the triggering magnets that are compatible with microelectromechanical systems (MEMS) processes for Wiegand sensors. Electrodeposited Co-rich alloys are potential candidates for easily sourced hard magnetic pole piece materials because they are high-throughput, high-energy permanent micro-magnets that are fully integrable and microfabrication-compatible.

Electrodeposited Co-rich magnets are promising candidates for MEMS applications such as actuators [15], energy generation and harvesting systems [16], magnetic field sources in magnetic sensors [17], and magnetic scales in position measurement systems [18]. The utilization of hard magnetic materials in MEMS applications requires the consideration of cost-effective advantages over commonly used hard magnetic materials such as sintered NdFeB. Various thin-film technologies, such as sputtering and molecular-beam epitaxy, could be used to deposit thin magnetic films such as Co-based and rare-earth alloys (such as SmCo) [19] according to various magnetic anisotropy configurations. However, due to the limitations of these vacuum processes, only relatively thin layers can be produced, leading to weak magnetic field strength at working distances; thus, they are typically not favored for their respective applications. The fabrication of thick magnetic layers could be realized using the atomic layer deposition [20] technique due to agglomeration of powder magnets. However, this process requires the use of rare-earth elements to achieve sufficiently high magnetic fields at a given distance due to lower packing density. As a candidate for alternative materials and deposition techniques, electrodeposited Co-based alloys [16] enable the cost-effective production of thick films and microfabrication-manufactured micro-magnets with a high degree of geometric freedom and precision.

Co-based alloy films produced by electrodeposition, including CoP, CoPt, CoNiP, CoMnP, CoWP, CoPtP, and CoNiMnP, hold great promise for use in MEMS [21,22,23]. The magnetic properties of these alloys are significantly influenced by the Co content, grain size, and crystalline texture, which in turn affects the magnetocrystalline anisotropy (MA) [24] of the deposited films. Additionally, electrodeposited Co-rich alloys exhibit hard magnetic properties which result from the isolation of fine Co grains by the nonmagnetic elements that are segregated along grain boundaries [25]. For example, binary CoP alloy yields a high magnetic strength for a low P content and better process control, and the direction of MA can be tailored through the parameters of electrodeposition. Alloying significantly alters the MA of tertiary Co-rich alloys such as the in-plane (IP)-type CoNiP alloy [18] and the out-of-plane (OP)-type CoMnP alloy [26]. The enhancement of the magnetic flux can be realized through the increase in the thickness of the films without altering intrinsic magnetic properties such as MA, magnetization, and energy product [27]. Moreover, Chen et al. have demonstrated a strategy of multilayer engineering to preserve the OP magnetic properties while enhancing the IP properties of CoMnP alloy [28,29]. However, when a thick Co-based hard magnetic layer is microfabrication-patterned and magnetized, shape anisotropy can be the dominant effect depending on the specific geometry. Therefore, developing and improving a specialty magnetization process that yields a well-controlled profile of induction lines at the working distances is challenging. In recent years, due to the rise of flexible material technology [30], magnetic materials have gradually developed towards curved-surface magnetism (curvilinear magnetism); correspondingly, emerging magnetization techniques such as origami magnetization have been found suitable for MEMS applications [31,32]. 

Flexible permanent magnets play a crucial role in the design of magnetically actuated micro-robots and microsystems that are manipulated through an external magnetic field [33,34]. Despite being important in various applications, flexible permanent magnets pose a significant challenge to achieving optimal magnetic performance due to their ultrathin geometries [35]. In combination with the microfabrication patterning processes [17,22] and the origami techniques [31], the resultant enhanced magnetic field strength and well-controlled field profile allow for the effective utilization of thin, flexible permanent magnets, which in turn appear to be promising for application in Wiegand sensors. This prompted us to investigate electrodeposited and origami-magnetized Co-rich micro-magnets and evaluate their feasibility as triggering magnets for Wiegand sensors.

The present study explores the potential of microfabrication-processed CoNiP micro-magnets through electrodeposition on flexible substrates, which can be utilized to control the profile of stray fields by means of origami magnetization. In-depth analysis of the effect of the triggering magnetic fields on the output pulse behavior of a micro-scale Wiegand sensor is also performed. As a benchmark, we adopted a conventional NdFeB magnet to trigger a Wiegand pulse, and comparative results are discussed.

## 2. Materials and Methods

### 2.1. Triggering Magnets and Material Characterization

In this research, a typical compact-sized NdFeB magnet (Figure 2a) was utilized as a benchmark for comparing the triggered Wiegand pulses. Commercial flexible copper-clad laminate (FCCL) were adopted as the flexible conductive substrates for the microfabrication of CoNiP micro-magnets. The consistency of the material properties of the FCCL substrates is excellent. The FCCL uses polyimide (PI) as its substrate and copper foil as the conductive layer, and its overall thickness is 50 μM.

The schematic in Figure 3 shows the detailed steps of the microfabrication process for fabricating CoNiP micro-magnets. Firstly, the as-received FCCL substrate was cropped to a size of 2.5 cm × 2.5 cm. The microfabrication of patterning and electrodeposition is based on an effective area defined by applying insulating Kapton tape to form a 2.0 cm × 2.0 cm area. The millimeter-scale pattern which defines the patterned areas to be electrodeposited was realized through sequentially applying Kapton tape and insulating varnish, as shown in Figure 3a. After baking and curing, the Kapton tape was removed to form an insulating varnish pattern with a height of several hundred microns, as depicted in Figure 3b. 

Considering the need for a uniform magnetic field profile in the horizontal (i.e., IP) direction, we adopted CoNiP alloy in the electrodeposition process, as shown in Figure 3c. The detailed bath composition and parameters of the electrodeposition are shown in Table 1. The electrodepositing process started with pre-treatment, including pickling for 20 s, water rinse, alkali wash for 5 s, water rinse, etc., followed by electrodepositing for 5 to 7 h with the parameters shown in Table 1. Note that the electrodepositing rate of CoNiP is about 20 μM per hour using the present parameters [18]. Compared with vacuum processes, the high depositing rate of electrodeposition can readily produce thick CoNiP deposits for providing sufficient magnetic field strength. After the electrodeposition process, the insulating varnish was removed using organic solvent (acetone). Finally, a pattern of electrodeposited Co-rich alloy was formed on FCCL substrate, as shown in Figure 3d. The top view of the design of patterned micro-magnets is shown in Figure 2b, with space (s) being 1.0 mm and width (w) of the micro-magnet being 2.0 mm.

The fabricated sample was inspected with an optical microscope (OM), model Edge 3.0 AM73915MZT (Dino-Lite, New Taipei City, Taiwan). A scanning electron microscope (SEM) was used to observe the surface morphology of the CoNiP film. The SEM model is HITACHI S-4800 (Tokyo, Japan). The SEM is equipped with an energy-dispersive spectrometer (EDS) for qualitative and semi-quantitative composition analysis. In order to avoid the charge accumulation of high-energy electrons on the CoNiP surface, a nano-layer of platinum was evaporation-deposited before observation. The working voltage of the SEM was 4~10 kV. An X-ray diffractometer (XRD) was used to analyze the crystal structure of the CoNiP film. The XRD is the TTRAX Ⅲ-type XRD produced by Rigaku (Tokyo, Japan), equipped with the light source of Cu K_α_. The operating voltage was 15 kV.

A superconducting quantum interference device magnetometer (SQUID) was used to measure the magnetic properties of the CoNiP sample for obtaining the hysteresis curves. The instrument is produced by Quantum Design in the United States, and the model is MPMS-3. The instrument can apply an external field in the range of ±70,000 Gauss (G) at the temperature range 1.9~400 K, and the measurable magnetization range is from 5 × 10^−8^ emu to 300 emu. In the measurement, the normal (i.e., OP) direction of the CoNiP coating and the horizontal (i.e., IP) direction of the CoNiP coating were characterized.

### 2.2. Origami Magnetization and Measurement of the Magnetic Field Profile

A C-shaped magnetizing head (Figure 4a) was designed to generate a uniformly applied magnetic field of 1.5 T for magnetization. Three configurations of magnetization, including flat, curved, and annular CoNiP samples, were realized through 3D-printed sample holders. The details of the 3D-printed sample holder for the curved magnetization are depicted in Figure 4b. The directions of the origami curling and the applied field are indicated in Figure 2b. During the curved magnetization process, the sample holder allowed the CoNiP sample to be clamped in the U-shaped groove of two holding fixtures. For annular magnetization, another 3D-printed sample holder, as shown in Figure 3c, was employed. A capacitive discharge pulse magnetizer (Ney Hwu Electrical Co., Ltd., Taipei, Taiwan) was used as the excitation source for magnetization, with a current setting of 500 A.

Figure 5 illustrates the apparatus and method for measuring the stray field profile of the hard magnets. The hard magnets were positioned on the moving platform composed of a servo-controlled stepper motor (08PTM-50M, Unice E-O Services Inc., Taoyuan City, Taiwan) which has a minimum step size of 2.5 μM and a repeatability of 5 μM. A Hall sensor tip of a Gaussmeter (F.W. Bell 5180, OECO LLC, Milwaukie, OR, USA) was then fastened on a manual stage that provides fine adjustment in y- and z-axis directions and θ rotation. The sensing directions of the Hall sensor include two IP directions and one OP direction (Figure 2b). The direction aligned to the Hall sensor moving direction (relative to the platform) was defined as the IP-X direction. The same method of defining directions was applied to the NdFeB magnet (Figure 5). Both the long side and short side of the NdFeB magnet were measured as depicted in Figure 2a. In the measurement process, first the Hall sensor sensing direction was adjusted and aligned with the principal direction of stray field so that the reading value of the Gaussmeter reached the local maximum. Next, the Hall sensor height in the z-axis was adjusted to search for a flight height providing a magnetic field strength of about 100 G. The minimum flight height of the Hall sensor was 0.5 mm if there was no 100 G field strength. Finally, the platform of the motorized linear stage was controlled to translate the hard magnet at an increment of 0.1 mm each step. Then, the field strength of each step and hence the profile of stray field could be measured at a specific flight height above the patterned CoNiP magnet or NdFeB magnet.

### 2.3. Output Voltage Measurement of the Triggered Wiegand Pulses

The experimental setup includes a USB data acquisition (DAQ) instrument (USB-6210, National Instruments (NI), Austin, TX, USA), Labview software (version 17.0f2) and a manual linear guide slide platform in order to measure the Wiegand out pulse (Figure 6). The Wiegand wire used in this experiment was taken from a commercially available Wiegand sensor, the WG631. Its specifications are detailed in Table 2 [36]. We extracted the Wiegand wire and made a custom Wiegand sensor using a sectioned Wiegand wire with a length of 9.0 mm and with a pick-up coil of 121-turn winding.

The Wiegand sensor is fixed in a clampable customized jig, and the entire jig is placed on an adjustable track to facilitate the adjustment and fixation of flight height. Note that the patterned CoNiP hard magnet was flattened when in use to trigger the Wiegand sensor. The relative motion between hard magnets and the Wiegand sensor was realized through a manual linear guide slide. To ensure the same initial status of Wiegand wire magnetization, it is necessary to flip the magnetic polarity of the Wiegand wire into an identical polarization using a magnet before triggering the Wiegand pulse. In the current study, positive pulses were generated, recorded, and compared. The flight height between the hard magnets and the Wiegand sensor was adjusted stepwise (e.g., ±0.5 mm per step), and the aforementioned Wiegand pulse measurement was repeated in order to find the optimal output voltage of the Wiegand pulse. As a benchmark, a commercially available NdFeB magnet was employed to trigger the Wiegand output pulse through the same method, and the results were compared and discussed.

## 3. Results and Discussion

### 3.1. NdFeB Magnets and Output Pulses

Figure 7 shows the measurement results of the spatial distribution of stray field intensity on the NdFeB magnet for triggering the Wiegand pulse. It is evident that the long side has distinguishable IP-X and IP-Y magnetic field strengths while the short side has a similar IP-X and IP-Y magnetic field strength, forming an equi-biaxial magnetic field in the IP directions.

Figure 8 depicts the output voltage of the Wiegand pulse triggered at different flight heights using the NdFeB magnet. The background noise is approximately in the range of 0.0005 V. In the case of the NdFeB magnet, it is evident from Figure 8a,b that the long-side triggering method yields a maximum output pulse of 0.023 V at a flight height of 25 mm, while the short-side triggering method produces a maximum output pulse of 0.014 V at a flight height of 20 mm. The reason for such an output voltage difference is because when the short side of the NdFeB is used for triggering pulses, the Wiegand wire experiences a multi-directional triggering field in the plane (that is, the plane formed by the moving Wiegand wire at a fixed flight height) where the magnetic field strengths in the IP-X and IP-Y directions are similar (Figure 7b). The influence of the equi-biaxial magnetic field leads to a disturbance to the magnetic reversal [10], making the overall output pulse voltage lower.

### 3.2. Microfabrication and Origami Magnetization of CoNiP Micro-Magnets

The magnetic studies of the CoNiP coating performed using SQUID suggest that the hard magnetic properties in the IP direction are better than those in the OP direction (Figure 9a,b). As shown in Table 3, there is little difference in the coercive force (H_c_) along the horizontal direction and the vertical direction. The coercive force is recorded as about 1100 Oe in both directions. On the other hand, there is a significant variation in the residual magnetization (B_r_). The residual magnetization in the IP direction is 4200 G, which is much higher than the residual magnetization in the OP direction, measuring 2500 G only. This remarkable difference in the residual magnetic flux density is ascribed to magnetocrystalline anisotropy as well as shape anisotropy of the coating.

To study the crystal structure of the CoNiP coating, the XRD technique was employed, and the recorded XRD pattern is shown in Figure 9c. The peaks identified from the XRD pattern indicate crystallization at the hexagonally close-packed (HCP) phase. The first peak in the XRD pattern is identified as Co HCP (100), indicating the horizontal MA which leads to better magnetic properties in the horizontal direction of the CoNiP coating. The evidenced low intensity of HCP (002) compared to that of HCP (100) indicates that the CoNiP coating has a significantly preferred orientation of HCP c-axis aligned in the horizontal direction. SEM results indicate that the CoNiP coating possesses nano-sized grain. The EDS results illustrate that the cobalt content is 87.1 ± 2.8 wt.%, the nickel content is 10.2 ± 0.6 wt.%, and the phosphorus content is 2.7 ± 0.2 wt.%, representing a Co-rich coating.

Figure 10a shows the photos of the patterned CoNiP magnet manufactured through the microfabrication process. The designed pattern can be completely formed, resulting in a flexible hard magnetic component. By inspecting the millimeter-scale pattern as depicted in Figure 10b, it is clear to see that a desired pattern of thick CoNiP coating was able to be realized through the microfabrication process using insulating varnish patterns.

The CoNiP sample was magnetized using the C-shaped magnetizing head by keeping the strip range against one open end of the C-shaped head. Once the magnetization process was complete, we performed quick measurements for the IP-X magnetic field strength using the Gaussmeter at nine positions above the CoNiP sample, as shown in Figure 11a. After such preliminary and screening measurements, the spatial distribution of stray field strength of the CoNiP samples was measured. As illustrated in Figure 11b–d, in all the 2 cm × 2 cm patterned CoNiP samples the upper part (regions i, ii, iii) and the lower part (regions vii, viii, ix) present two opposite polarities through the plus and minus signs convention. The highest magnetic field strength obtained for each sample falls in the upper part of the sample, which is also the strip region used to trigger the Wiegand pulse. For the strip regions, the maximum magnetic field strength of the flat magnetized sample is about −60 G and the minimum is −0.2 G, as shown in Figure 11b. This result indicates that the difference in magnetic field strength caused by the fringe effect can reach hundreds of times. On the other hand, the maximum magnetic field strength of the curved CoNiP sample by origami magnetization is about −100 G and the minimum is about −50 G. The difference in magnetic field strength caused by the fringe effect drops to about 2 times, as depicted in Figure 11c. This difference can be further reduced to less than 2 times by annular origami magnetization, as shown in Figure 11d.

### 3.3. CoNiP Micro-Magnets and Output Pulses

Figure 12 depicts the outcomes of the spatial distribution of stray field intensity measured from the CoNiP samples used to trigger Wiegand pulses in this study. Note that the flat magnetized CoNiP sample can hardly trigger any Wiegand pulses. It is clear to see that the field strength in two IP directions (i.e., IP-X and IP-Y) has a significant difference and the OP field strength is much weaker compared with any of the IP field strengths. Note that the flight height of the Hall sensor probe tip of the Gaussmeter is 0.5 mm when characterizing the CoNiP samples. The Wiegand pulse triggering experiment was carried out using the regions indicated by the red dashed arrows in Figure 12 (at a flight height of around 0.5 mm). On comparing the results of the patterned CoNiP and the commercial NdFeB magnets, one can conclude that a single-axis stray field in the plane where Wiegand wire flies is achieved through the formation of magnetocrystalline anisotropy and shape anisotropy of the patterned CoNiP magnet. It is also found that origami magnetization can effectively control the magnetic field strength and profile, which is difficult to tackle for bulk and rigid magnets due to the fringe effect. It is worth noting that the surface of the CoNiP micro-magnets does not present complete evenness due to significantly accumulated residual stress of the thick deposits (Figure 10). This is the reason for the ripple-like field profiles seen in Figure 12. Chen et al. have demonstrated that multilayer engineering using thin copper interlayers can efficiently resolve the residual stress issue and also reserve hard magnetic properties of the electrodeposited micro-magnets [28,29]. Related research work is in progress in our research group for further novel MEMS design purposes and will be published in the near future.

Figure 13a represents the results of the Wiegand pulse triggering test conducted using the strip region of the patterned CoNiP magnet, and it reveals that a maximum pulse voltage of about 0.027 V is achieved at a flight height of 0.5 mm. Note that the output pulse voltage decreases when the flight height deviates from the optimal value [6]. The above results suggest that the external magnetic field having a prominently uniaxial stray field is key to maximizing the output voltage when the principal axis of stray field aligns well with the Wiegand wire. In this way, the three-dimensional tensor of the stray field distribution can be a precise fit for the wire-shaped Wiegand element (as illustrated in Figure 12) to trigger a magnetization reversal of great magnitude.

The results of the optimal output pulses are summarized in Figure 13b for the comparison between the patterned CoNiP micro-magnet and the NdFeB magnet. The peak voltage values of the optimal pulses triggered by the patterned CoNiP and the long side of NdFeB magnets are similar (both around 0.025 V). On the other hand, the annular-magnetized CoNiP magnet produces a peak voltage value of 0.018 V, and the NdFeB short-side triggered output pulse is 0.014 V. Moreover, all measurement results indicate that the full width at half maximum (FWHM) of all the triggered pulses is in the range of 10 μs to 30 μs, which is consistent with the specification as show in Table 2, thus representing a typical Wiegand pulse.

## 4. Conclusions

In this study, electrodeposited CoNiP hard magnets on flexible substrates were successfully patterned using a microfabrication technique. The patterned CoNiP magnets can be origami magnetized to realize an engineered stray field. This study has illustrated the effective triggering of Wiegand pulses using the pole piece of CoNiP micro-magnets. The results of the study indicate that the output voltage of the triggered pulse using the patterned CoNiP magnets is comparable to that acquired using rare-earth magnets. Additionally, the use of CoNiP micro-magnets allows for a significant reduction in both the volume of the Wiegand sensors and the EMI issue.

In the future, the MEMS process-compatible electrodeposited Co-rich micro-magnets could facilitate the application designs of Wiegand sensors. The positive and negative Wiegand pulses could be modulated directly through the specialty patterned and magnetized micro-magnets. Further parameters of Wiegand sensors, including the hard layer polarization, length of Wiegand wire (i.e., the ratio of wire length to wire radius), and shape (through the alternative materials that can realize the Wiegand effect), could be efficiently utilized because of the breakthrough of the controlled magnetic field profile of the triggering magnets. Novel Wiegand sensor-integrated devices such as self-powered positioning systems and rectifier-free energy harvesters will become possible through further research and development.

## Figures and Tables

**Figure 1 sensors-23-06043-f001:**
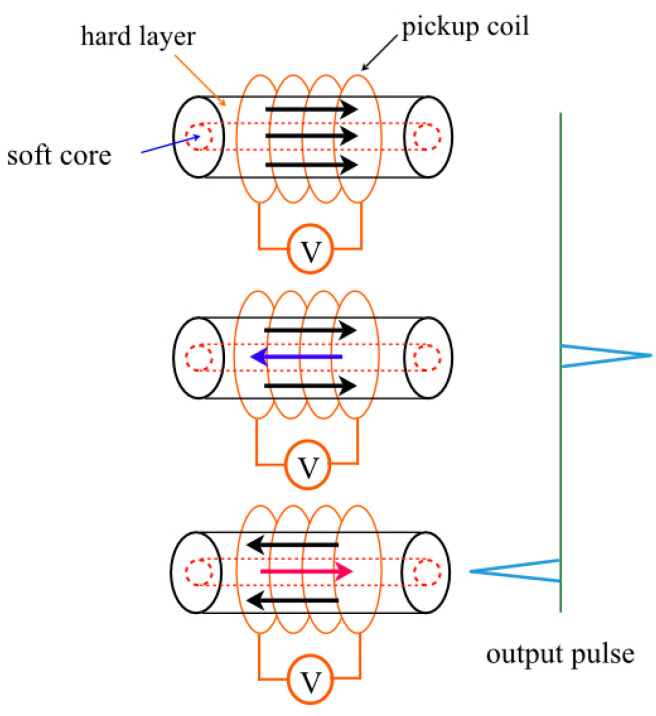
Schematic representation of the Wiegand effect. The red and blue arrows illustrate the magnetization reversal of soft core, which accordingly induces a positive and a negative output pulses in the pickup coil.

**Figure 2 sensors-23-06043-f002:**
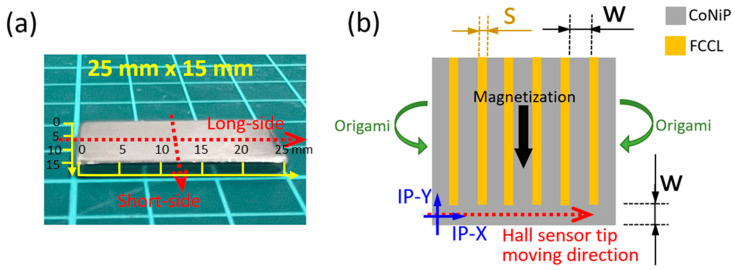
(**a**) The NdFeB magnet and (**b**) schematic drawing of the patterned CoNiP magnet showing the geometries and the definitions of the measurement directions for the magnets. The blue arrows illustrate the Hall sensor sensing directions (out-of-plane (OP) being normal to in-plane (IP) directions), while the red-dot arrows indicate the moving tracks of the Hall sensor tip. In (**b**), the directions of applied field for magnetization (black arrow) and sample curling for origami (green arrows) are illustrated accordingly.

**Figure 3 sensors-23-06043-f003:**
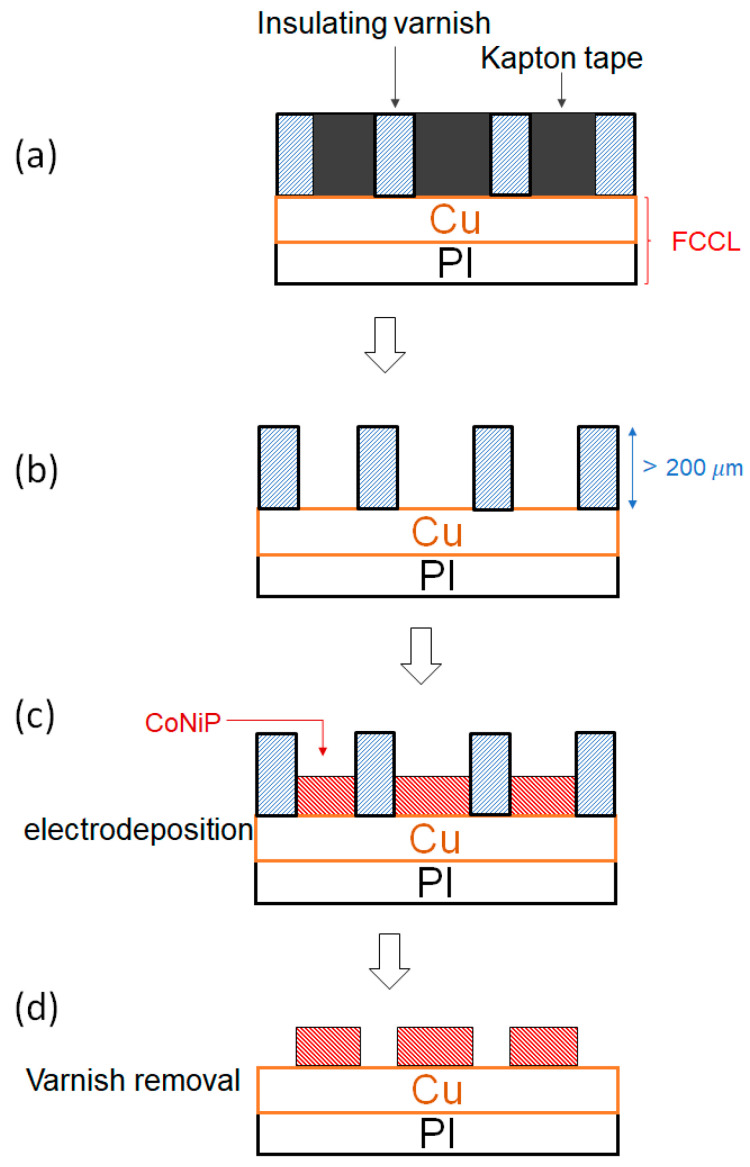
Schematic representation of the microfabrication process: (**a**) Applying the Kapton tapes and subsequently applying the insulating varnish on the FCCL substrate; (**b**) Removing Kapton tapes after curing of the insulating varnish; (**c**) Electrodepositing CoNiP micro-magnets; (**d**) Removing the insulating varnish to form the patterned CoNiP micro-magnets.

**Figure 4 sensors-23-06043-f004:**
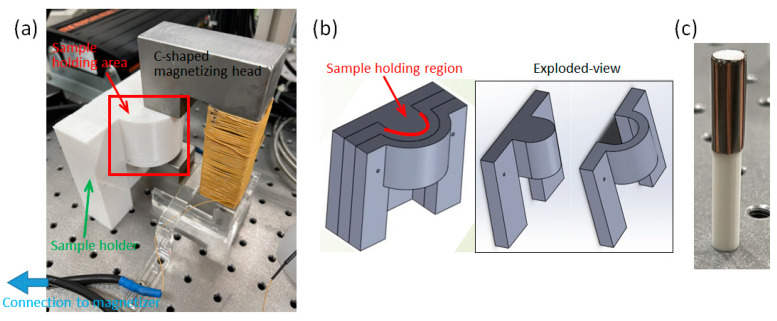
Details of the magnetization setups: (**a**) the C-shaped magnetizing head and the 3D-printed sample holder; (**b**) 3D CAD model of the sample holder and its exploded view; (**c**) sample holder along with a patterned CoNiP sample for annular magnetization.

**Figure 5 sensors-23-06043-f005:**
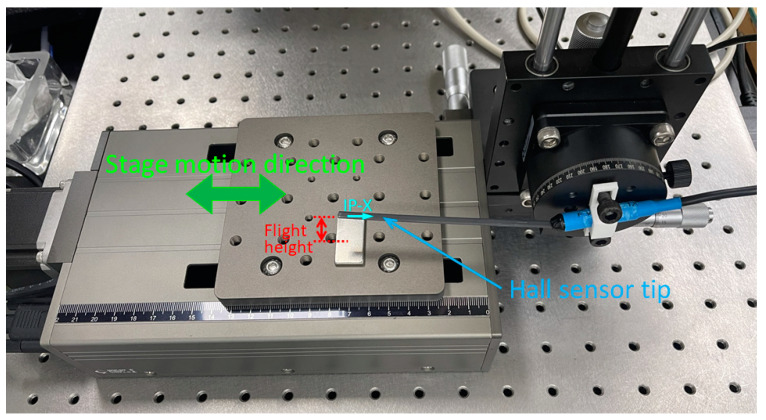
Apparatus composed of a Hall sensor, a y–z-θ manual stage, and a motorized linear stage for measuring the magnetic flux density profile of the magnets.

**Figure 6 sensors-23-06043-f006:**
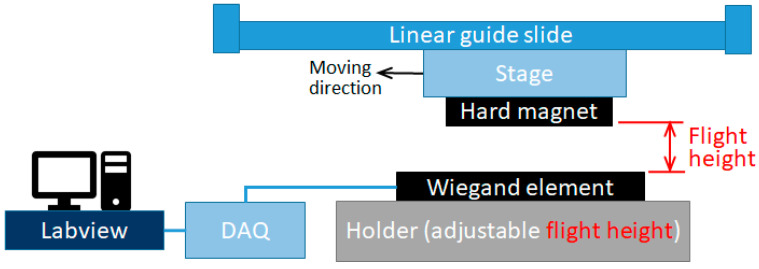
Schematic drawing showing the setup for Wiegand pulse measurement.

**Figure 7 sensors-23-06043-f007:**
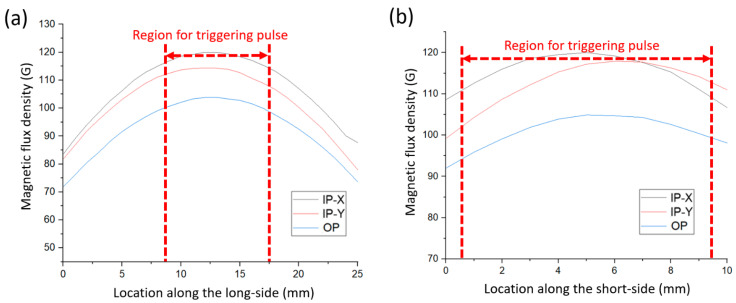
Measurement results of the magnetic flux density profiles of the NdFeB magnets along the long side (**a**) and the short side (**b**).

**Figure 8 sensors-23-06043-f008:**
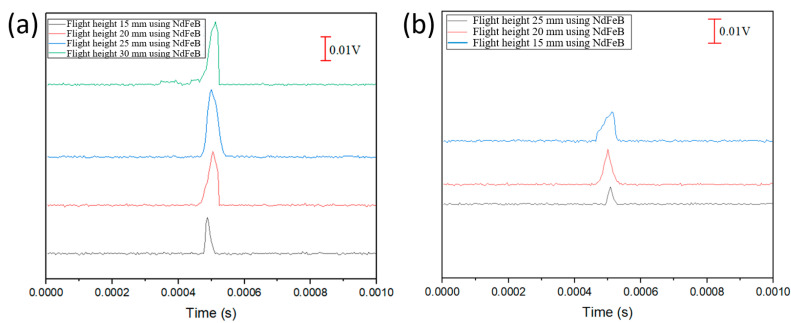
Output pulses (presented in stacked format) as a function of flight height: (**a**) triggered using the long side of the NdFeB magnet; (**b**) triggered using the short side of the NdFeB magnet.

**Figure 9 sensors-23-06043-f009:**
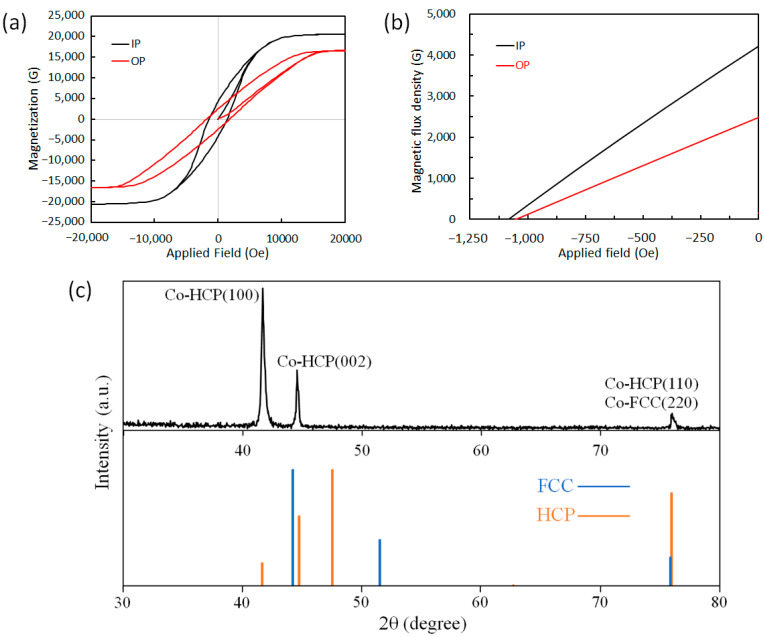
Materials characterization results of CoNiP magnets showing the SQUID-acquired M-H curves (**a**), second quadrant of B-H curves (**b**), and XRD pattern along with the information of JCPDS (Joint Committee on Powder Diffraction Standards) cards (**c**) for cobalt.

**Figure 10 sensors-23-06043-f010:**
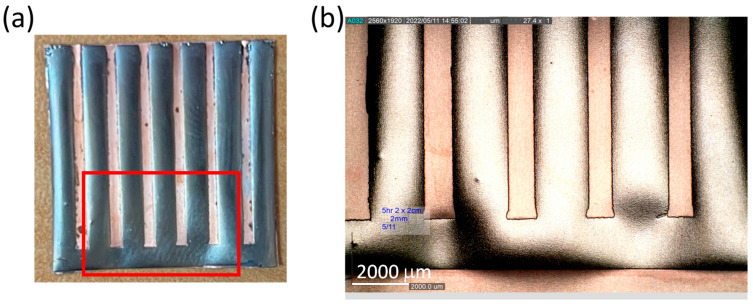
(**a**) Photo of the patterned CoNiP magnet and (**b**) the zoomed-in image of the red rectangular area in (**a**).

**Figure 11 sensors-23-06043-f011:**
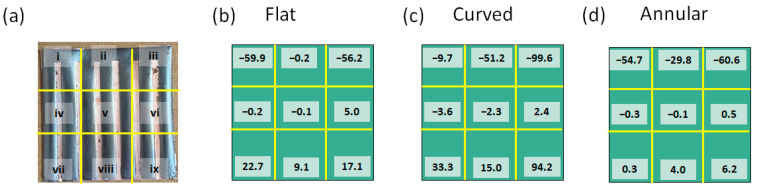
Brief measurement of the stray field intensities above the patterned CoNiP magnets: (**a**) definition of the nine (9) measurement locations marked as i–ix; and the stray field intensities (unit in Gauss) in the nine locations of the CoNiP magnets after (**b**) flat magnetization, (**c**) curved magnetization, and (**d**) annular magnetization.

**Figure 12 sensors-23-06043-f012:**
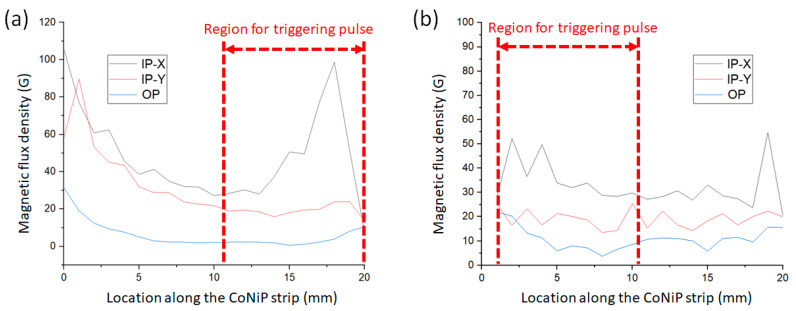
Measurement results of the magnetic flux density profiles along the strip of the patterned CoNiP magnets after (**a**) curved magnetization and (**b**) annular magnetization.

**Figure 13 sensors-23-06043-f013:**
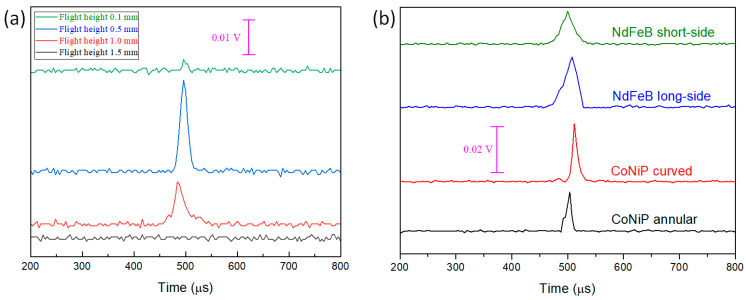
(**a**) Output pulses (presented in stacked format) as a function of flight height triggered using the strip region of the patterned CoNiP magnet by curved magnetization. (**b**) Comparison of the optimal output pulses (presented in stacked format) triggered by the NdFeB and the patterned CoNiP magnets.

**Table 1 sensors-23-06043-t001:** The bath compositions and depositing parameters for the electrodeposition of the CoNiP magnet.

Stoichiometry of Magnetic Particles	Bath Composition	Concentration (Mole/L)
CoNiP	CoCl_2_·6H_2_O	0.2
	NiCl_2_·6H_2_O	0.2
	NaH_2_PO_2_·H_2_O	0.3
	H_3_BO_3_	0.4
	NaCl	0.7
Electrodeposition	Deposition Parameter	Settings
Galvanic deposition	Bath volume	400 mL
	pH value	4.2
	Current density (CD)	20 mA/cm^2^
	Temperature	25 °C
	Agitation	Air bubbling

**Table 2 sensors-23-06043-t002:** Properties of the Wiegand element acquired from the commercial WG631 Wiegand sensor.

Properties of Wiegand Element	External Field (G)	Pulse FWHM (μs)	Frequency (kHz)	Working Temperature (°C)
Nominal minimum	55	10	0	−40
Nominal maximum	120	30	5	125

**Table 3 sensors-23-06043-t003:** Characterization results showing the hard magnetic properties in the in-plane and out-of-plane directions of the electrodeposited CoNiP magnets.

Direction of Characterization	Hci (Oe)	Ms (G)	Hc (Oe)	Br (G)	(BH)max (MGOe)
In-plane	1412	18,050	1083	4218	1.16
Out-of-plane	1803	14,550	1050	2482	0.65

## Data Availability

All data that support the findings of this study are included within the article.

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
