# Peer review of "Triggering Magnets for Wiegand Sensors: Electrodeposited and Origami-Magnetized CoNiP Micro-Magnets"

_sensors, 2023, doi:10.3390/s23136043_

Round 1
Reviewer 1 Report
1. In third paragraph, “Moreover, typical diameter of the Wiegand wire is 0.25 mm”, please add relevant references which supported this statement.
2. In Figure 9(b), there are some Chinese descriptions in the picture, which is not standardized enough.
3. Full name of some abbreviations are not shown (e.g. DAQ). The authors should avoid this situation.
4. In Figure 8(b), the image format of XRD is not consistent, please modify the similar situation in this article.
5. The surface of CoNiP magnet is unevenness, will this have an impact on its performance?
Reviewer 2 Report
Wiegand sensors are attractive as they are self-powered and could be used for sensing and energy harvesting simultaneously. However, the design and adoption of external triggering fields for the Wiegand sensor are limited by the wire-shaped sensing element of the Wiegand wire. In the manuscript, patterned CoNiP hard magnets were electrodeposited on flexible substrates through microfabrication. Origami magnetization was utilized to control the resultant stray fields, and hence the pole piece of CoNiP magnets can successfully trigger the output of Wiegand pulse with reduced weight and EMI issues. The improvement made by the authors could be important for the development of wearable electronics, below are my comments:
1、The fabrication process of the patterned CoNiP magnet is not easy to follow. I suggest the authors schematically draw the step-by-step process in Fig.2.
2、I think it is necessary to indicate the start point of the location along the long-side and short-side (Fig.6) in Fig.2 or Fig.4.
3、In line 301-302,the authors remark that the hard magnetic properties in the IP direction are better than those in the OP direction. Therefore, it is better to plot full M-H loop in Fig.8(a).
Reviewer 3 Report
The manuscript describes developments for Wiegand sensors. I think the Introduction part could be written in more detail concerning that for what sensor applications are the outlined sensors suitable.
The authors claimed that 5 to 7 hours were necessary for the electrodeposition. It seems too long for electrodeposition of small magnetic particles.
I suggest to write "stoichometry of magnetic particles" instead of "bath formulation" in Table 1.
In caption of Figure 6 modify to "along the long side (a) and the short side (b)".
In the discussion of results of XRD studies what does the abbreviation HCP mean?
Define what is S and N polarity.
Concerning the FWHM is it advantageous if the peaks are narrow?
English is acceptable but the entire text should be checked again due to some mistyping errors.
Round 2
Reviewer 1 Report
/
/